# Isolation of Actinobacteria from Date Palm Rhizosphere with Enzymatic, Antimicrobial, Antioxidant, and Protein Denaturation Inhibitory Activities

**DOI:** 10.3390/biom15010065

**Published:** 2025-01-05

**Authors:** Maria Smati, Amina Bramki, Fatima Zohra Makhlouf, Rihab Djebaili, Beatrice Farda, Fatima Zohra Abdelhadi, Nahla Abdelli, Mahmoud Kitouni, Marika Pellegrini

**Affiliations:** 1Higher National School of Biotechnology Taoufik Khaznadar, Nouveau Pôle Universitaire Ali Mendjeli, BP. E66, Constantine 25100, Algeria; makhlouf_f.zohra@umc.edu.dz (F.Z.M.); abdelhfatima@gmail.com (F.Z.A.); nahlaimane35@gmail.com (N.A.); 2Laboratory of Microbiological Engineering and Applications, University of Brothers Mentouri, Constantine 1, Chaâbat Erssas Campus, Ain El Bey Road, Constantine 25000, Algeria; mahmoudkitouni@yahoo.fr; 3Laboratory of BioEngineering, Higher National School of Biotechnology Taoufik Khaznadar, Nouveau Pôle Universitaire Ali Mendjeli, BP. E66, Constantine 25100, Algeria; a.bramki@ensbiotech.edu.dz; 4Department of Life, Health and Environmental Sciences, University of L’Aquila, Coppito, 67100 L’Aquila, Italy; beatrice.farda@graduate.univaq.it (B.F.); marika.pellegrini@univaq.it (M.P.)

**Keywords:** actinobacteria, *Phoenix dactylifera* L., biomolecules, Ghardaia, Algeria

## Abstract

Arid ecosystems constitute a promising source of actinobacteria producing new bioactive molecules. This study aimed to explore different biological activities of actinomycetes isolated from the rhizosphere of *Phoenix dactylifera* L. in the Ghardaia region, Algeria. A total of 18 actinobacteria were isolated and studied for their enzymatic and antimicrobial activities. All isolates shared cellulase and catalase activity; most of them produced amylase (94%), esterase (84%), lecithinase and lipoproteins (78%), caseinase (94%), and gelatinase (72%). The isolates could coagulate (56%) or peptonize (28%) skim milk. Overall, 72% of the isolates exhibited significant antibacterial activity against at least one test bacteria, while 56% demonstrated antifungal activity against at least one test fungi. Based on enzyme production and antimicrobial activity, isolate SGI16 was selected for secondary metabolite extraction by ethyl acetate. The crude extract of SGI16 was analyzed using DPPH and BSA denaturation inhibition tests, revealing significant antioxidant power (IC_50_ = 7.24 ± 0.21 μg mL^−1^) and protein denaturation inhibitory capacity (IC_50_ = 492.41 ± 0.47 μg mL^−1^). Molecular identification based on *16S rDNA* analysis showed that SGI16 belonged to the genus *Streptomyces*. The findings highlight that date palms’ rhizosphere actinobacteria are a valuable source of biomolecules of biotechnological interest.

## 1. Introduction

To meet the growing demand for new compounds in various industrial and agricultural sectors, as well as to counteract antibiotic-resistant pathogens, research and industries are exploring new microorganisms in diverse and understudied environments [1]. Among the microbes of interest, actinobacteria are great producers of various biomolecules [2,3].

Actinobacteria, also known as actinomycetes, are Gram-positive filamentous bacteria, sporigenous, with high G + C content in their DNA (around 70%) [4,5]. Bioactive metabolites derived from actinobacteria account for approximately 70% of natural compounds currently used clinically [6]. With these metabolites, antiviral, antifungal, antimalarial, antibacterial, immunosuppressive, antitumor, enzyme inhibitor, antioxidant, anti-inflammatory, and cytotoxic drugs have been described [7]. Additionally, actinobacteria secrete a wide range of enzymes with significant industrial relevance [8]. Amylase, for example, is used in the food, textile, and paper industries [9]. Cellulase, on the other hand, is employed in the production of biofuels, textiles, paper pulp, and detergents, as well as in the food and animal feed industries [10,11]. Lipase is essential in the detergent, food, and pharmaceutical industries [12]. Lastly, proteases are widely used in the food, pharmaceutical, leather, detergent, and photography sectors [13]. 

Actinobacteria are present in both terrestrial and aquatic habitats. This lineage is widely found in nutrient-rich rhizospheres, accounting for high percentages of microbial communities and where they produce various agricultural bioactive compounds with plant growth-promoting properties [14,15,16,17]. Given their cellular resistance and wide-range metabolic adaptation, they can also colonize extreme environments [18], where they can biosynthesize a wide variety of novel natural bioactive compounds [19]. Among the extreme environments, the Algerian Sahara is part of the world’s largest hot desert, occupying almost 90% of the country’s surface [20]. Its climate is characterized by low and irregular precipitation, high temperatures, intense sunlight, and high evaporation [21].

Only limited research has been focused on date palms’ rhizosphere actinobacteria in these desertic soils. Some studies have explored the biodiversity and antimicrobial activity of actinobacteria in Tamenrasset [22], as well as their distribution in the rhizosphere of date palms sensitive and resistant to *Fusarium* wilt in Adrar [23]. Thus, the main objective of this research is to isolate bacteria belonging to the Actinobacteria phylum from the rhizosphere of date palms (*Phoenix dactylifera* L.) in the Ghardïa desert of Algeria (an ecosystem that, to the best of our knowledge, has not been previously exploited) and to evaluate their potential to produce biologically active substances.

Based on the rhizosphere containing organic matter from an arid climate, which has not been the subject of microbiological and biotechnological studies, we hypothesized that this environment is rich in microorganisms producing new bioactive compounds. To test this hypothesis, we collected rhizosphere samples from the desert date palm in Ghardaia, from which we highlighted the ability of culturable actinobacteria to produce enzymes and antibacterial and antifungal agents. Based on different tests, a single isolate was selected to extract its secondary metabolites and study their antioxidant and protein denaturation inhibitory activities. The molecular identification of this isolate was also carried out based on the phylogenetic analysis of *16S rDNA*.

## 2. Materials and Methods

### 2.1. Area Study, Sample Collection, and Soil Physicochemical Characterization

A rhizosphere sample from date palm (*Phoenix dactylifera* L.) was obtained from the Noumérat region of Ghardaïa city in the Sahara Desert, located in north-central Algeria (32°29′27″ N; 3°40′24″ E). Sampling was carried out on three different points. Rhizosphere soil was collected by gently shaking loose soil from the roots and scraping firmly, adhering soil from the root surface into sterile containers [24,25]. The collected samples were kept refrigerated. The samples for the isolation of actinobacteria were processed as soon as they were brought to the laboratory. The electrical conductivity, pH, organic matter, and moisture content of the soil samples were measured according to established methods [24,26,27].

### 2.2. Isolation of Cultivable Actinobacteria

The soil was dried at room temperature for one week. Filamentous actinobacteria were isolated using the suspension-dilution method. One gram of soil was suspended in 9 mL of sterile physiological water (0.9%) and vigorously shaken using a vortex. A series of decimal dilutions was performed up to a dilution of 10^−5^. Inoculation was carried out by spreading 0.1 mL of each dilution onto the surface of two culture media, SCA (Starch Casein Agar) [28] and ISP_2_ (International *Streptomyces* Project 2) [29]. To prevent the development of other bacteria and fungi, the culture media were supplemented with nalidixic acid (75 µg mL^−1^) and amphotericin B (25 µg mL^−1^). The plates were incubated at 30 °C and observed daily for a period of 7 to 21 days. The total number of colonies exhibiting the morphological characteristics of mycelial actinobacteria (presence of substrate mycelium and, very often, aerial mycelium) was expressed in CFU g^−1^. These colonies were purified and preserved on the same isolation media in slant tubes at 4 °C for further studies.

### 2.3. Enzymatic Activities of Actinobacteria

Various enzymatic tests on agar media were performed to detect the production of extracellular hydrolases by the actinobacteria isolates. The different tests are described as follows:Cellulolytic and amylolytic activities: Actionobacteria isolates were plated on ISP_2_ agar medium containing 1% Carboxy Methyl Cellulose (CMC) and on nutrient agar supplemented with 1% soluble starch. After incubation at 30 °C for 7 days, the cultures were covered with Lugol solution. The presence of clear halo around the colonies indicated cellulase and amylase activity [30].Esterase (lipase) activity: Actinobacteria were grown on Sierra’s medium [31] supplemented with tween 80 (1%). After 7 days of incubation at 30 °C, the appearance of an opaque halo around the colonies indicated esterase activity.Lecithinase and lipoproteinase: Isolates were streaked on 10% egg yolk agar and incubated at 30 °C for 7 days and two enzymes were tested: (1) lecithinase, with the appearance of an opaque, yellowish-white pearly halo with a clear edge under the colony or at its limit; (2) lipoproteinase, with the appearance of a clear zone around the culture [32].Caseinase: Protease activity was detected by inoculating actinobacteria onto milk nutrient agar containing 5% skimmed milk. After incubation for 7 days at 30 °C, a clear halo around the colonies indicated caseinase activity [33].Gelatinase: Actinobacteria were inoculated into tubes containing nutrient gelatin and incubated at 30 °C for 21 days. The tubes were then refrigerated for one hour. Solidification of the gelatin indicated that it had not been hydrolyzed, while liquid gelatin indicated the presence of gelatinase [33].Peptonization and coagulation of milk: Isolates were inoculated into tubes containing sterile skimmed milk and incubated at 30 °C. Regular observations over 14 days were performed to record the coagulation and peptonization of milk induced by each isolate [34].Catalase: A colony of each isolate was placed on a slide containing a drop of 10% hydrogen peroxide. The presence of catalase was indicated by the appearance of air bubbles due to the release of O_2_ gas [35].

### 2.4. Antimicrobial Activity

The antimicrobial activity of actinobacteria was tested against a panel of pathogenic microorganisms, including three Gram-positive (*Enterococcus faecalis* ATCC 29212, *Bacillus spizizenii* ATCC 6633, and *Staphylococcus aureus* ATCC 25923) and four Gram-negative bacteria (*Pseudomonas aeruginosa* ATCC 27853, *Escherichia coli* ATCC 25922, *Klebsiella pneumoniae* ATCC 700603, and *Salmonella typhimurium* ATCC 14028), one yeast (*Candida albicans* ATCC 10231), and four filamentous fungi (*Fusarium oxysporum* CIP 62572, *Aspergillus niger* MH109542, *Penicillium* sp., and *Alternaria* sp.). Suspensions of these microorganisms were prepared according to the protocols of Bramki et al. [36] and Nemouchi et al. [37]. Actinobacteria strains were inoculated on ISP_2_ medium and incubated for 14 days. Agar cylinders (6 mm diameter) from well-developed and well-sporulated cultures were cut out and placed on the culture media (Mueller Hinton for bacteria and Sabouraud for fungi) previously inoculated with the test germs. Petri dishes were kept at 4 °C for 4 h to enable proper diffusion of bioactive metabolites [21]. Then, inhibition diameters were measured after 24 to 48 h of incubation at 37 °C for bacteria and yeast and for 48 to 72 h at 28 °C for filamentous fungi.

### 2.5. Selected Isolate Bioactive Molecules Extraction

Well-developed cultures of the SGI16 isolate (10 days old) were fragmented into small pieces and completely covered with ethyl acetate. Maceration was carried out twice in order to recover the maximum of produced bioactive molecules. The crude extracts were filtered through Whatman N° 01 paper (11 µm) and then concentrated to dryness using a rotary evaporator (Heidolph, Germany) [38].

#### 2.5.1. Evaluation of Antioxidant Activity

The antioxidant effect of the SGI16 isolate extract was assessed by measuring the DPPH (1,1-diphenyl-2-picryhydrazyl, Sigma-Aldrich, St. Louis, MO, USA) (radical scavenging capacity. A volume of 2 mL of each extract concentration was mixed with 1.6 mL of 0.002% methanolic DPPH solution. The mixture was incubated at room temperature in the dark for 30 min. Ascorbic acid was used as a positive control and the absorbance was measured against a blank at a wavelength of 517 nm. The percentage DPPH free radical inhibition was calculated according to the equation:$$DPPH\;scavenging\;effect\;\left(\%\right)\;=\;\frac{{Absorbance}_{Control}-{Absorbance}_{sample}}{{Absorbance}_{Control}}\times 100$$

The results were expressed as IC_50_ values (μg mL^−1^), representing the concentration required to achieve 50% inhibition [39].

#### 2.5.2. Evaluation of Protein Denaturation Inhibitory Capacity

The ability of the extract to inhibit protein denaturation was determined using the method of Kar et al. [40] with slight modification. In a 96-well microplate, 100 µL of extract was added to 100 µL of 0.2% BSA (Bovine Serum Albumin, Sigma-Aldrich, St. Louis, MO, USA), and the mixture was then incubated at 72 °C for 20 min. After cooling, turbidity was measured at 660 nm against a blank prepared from 100 µL of Tris-HCl buffer (0.05 M, pH 6.6) and 100 µL of the SGI16 isolate extract. A negative control was prepared with 100 µL of BSA and 100 µL of ultrapure water, while diclofenac was used as a positive control. The inhibition percentage was calculated using the following formula:$$\%Inhibition=\frac{{Absorbance}_{Control}-{Absorbance}_{sample}}{{Absorbance}_{Control}}\times 100$$

#### 2.5.3. Molecular Identification and Phylogenetic Analysis of the SGI16 Isolate

The *16S rRNA* gene barcoding was performed on the most promising isolate, SGI16, by The Environmental Sciences Section, Department MeSVA, University of L’Aquila and BMR Genomics, Padua, Italy. DNA was amplified by direct PCR using universal bacterial primers (27F/1492R) and then sequenced. The obtained sequence was analyzed by Finch TV software version 1.4.0 (Geospiza, Inc.; Seattle, WA, USA; https://www.softpedia.com/get/Science-CAD/FinchTV.shtml, accessed on 1 September 2024) and compared with those present in the EZBioCloud database [41] (https://www.ezbiocloud.net/, accessed on 2 September 2024). The sequences were aligned using the Clustal X 2.0.12 program [42]. A rooted phylogenetic tree was constructed with MEGA version 11 [43]. The distance matrix was computed using the neighbor-joining method [44], which is based on the Kimura two-parameter model [45]. Tree topology was assessed using bootstrap analysis with 1000 replicates [46]. This analysis involved 12 nucleotide sequences. All positions containing gaps and missing data were eliminated (complete deletion option). There were a total of 1172 positions in the final dataset. The SGI16 sequence is available in GenBank under accession number PQ240116.

### 2.6. Statistical Analysis

All analyses were carried out in triplicate and the experimental data were reported as means ± standard deviation. Statistical analysis was carried out using XLSTAT software 2014.5.03 (Addinsoft, New York, NY, USA). Significant differences were determined at *p* ≤ 0.05 by one-way analysis of variance (ANOVA) followed by Tukey’s HSD post hoc test.

## 3. Results

### 3.1. Soil Physicochemical Characterization and Isolation of Actinobacteria

The physicochemical characteristics of the rhizosphere are listed in Table 1.

The filamentous colonies of actinobacteria ranging in size from 1 to 10 mm, with a powdery, rough, rugged, colored, or uncolored appearance, and often embedded in the agar, with or without aerial mycelium, were counted. According to the obtained results, the total number of actinomycetes isolated from the sample on the SCA medium (7.47 × 10^5^ ± 1.02 CFU g^−1^) was significantly higher than that obtained on the ISP_2_ medium (3.15 × 10^5^ ± 0.26 CFU g^−1^). Therefore, one gram of rhizosphere contains approximately 7.47 × 10^5^ filamentous actinobacteria. After purification, 18 morphologically different actinomycete isolates were selected. The isolates were coded based to their origin and the isolation media as follows: SGA4, SGA6, SGA7, SGA8, SGA9, SGA10, SGI2, SGI5, SGI6, SGI7, SGI8, SGI13, SGI16, SGI18, SGI19, SGI22, SGI24, and SGI25.

### 3.2. Enzymatic Activities

The different enzymatic activities of the obtained actinomycetes isolates are presented in Table 2.

All the obtained isolates showed positive cellulolytic activity. Most of the isolates (94%) exhibited amylase production and possessed enzymes capable of degrading lipids, such as esterase (83%), lecithinase, and lipoproteinase (78%). Regarding gelatin hydrolysis, 72% of actinobacteria were positive. Slightly over half (56%) of these bacteria showed peptonization of skimmed milk, while the remaining isolates (28%) caused coagulation.

### 3.3. Antimicrobial Activity

Most actinomycetes showed broad-spectrum antimicrobial activities against different pathogenic microorganisms. The positive results of the antibacterial activities are reported in Figure 1.

Thirteen actinomycete isolates (72.22%) of the eighteen studied produced metabolites with antibacterial activity against at least one of the seven test bacteria. Significant antibacterial activity was observed against *B. spizizenii* (28 ± 1.00 mm), followed by *S. aureus* (19 ± 1.00 mm) and *E. faecalis* (17 ± 1.00 mm). The isolates SGI13, SGI16, and SGA9 exhibited significant activity against *S. typhimurium*, with inhibition zones ranging from 10 to 15 mm. In addition, isolates SGI13, SGI16, SGA10, SGA7, and SGA8 demonstrated significant activity against *K. pneumoniae*, with inhibition zones ranging from 8 to 13 mm. Inhibition zones of 11 mm and 12 mm were observed for isolates SGI16 and SGA9 against *E. coli*. In contrast, no activity was detected against *P. aeruginosa*.

The positive results of the antifungal activities are reported in Figure 2.

Ten out of eighteen actinobacterial isolates (55.55%) exhibited antifungal activity against at least one of the five tested phytopathogenic or clinical strains. Among them, only one isolate (SGI16) inhibited *A. niger*, while two others (SGI16 and SGA9) were active against *F. oxysporium.* Ten isolates (55.55%) showed significant activity against *Penicillium* sp., and three isolates (SGI1, SGA9, and SGI19) were effective against *Alternaria* sp. Only one actinobacterium (SGA9) was significantly active against *C. albicans*. The maximum inhibition zones of 30.33 ± 0.57 and 31 ± 1.00 mm in diameter were produced by isolates SGA6 and SGI16 against *Penicillium* sp. and *Alternaria* sp., respectively. Furthermore, isolate SGI16 showed significant activity against *F. oxysporium*, with an inhibition zone of 25 ± 1.00 mm, compared to other isolates.

### 3.4. Selected Isolate’s Bioactive Molecule Extraction

The isolate SGI16, identified as the best producer of enzymes and antimicrobial agents, was selected to study its antioxidant and protein denaturation inhibitory activities. Indeed, in addition to its ability to produce seven enzymes (cellulase, amylase, catalase, esterase, lipoproteinase, caseinase, and gelatinase), it also exhibits activity against Gram-positive bacteria such as *S. aureus* and *E. faecalis*, as well as the Gram-negative *S. typhimurium*, *E. coli*, and *K. pneumoniae*, with inhibition zone diameters ranging from 10.66 ± 0.57 mm to 13.33 ± 0.57 mm. Moreover, it effectively acts against all tested molds, including *A. niger*, *F. oxysporum*, *Penicillium* sp., and *Alternaria* sp., with inhibition zone diameters ranging from 12 ± 0.00 mm to 31 ± 1.00 mm. This isolate was cultivated on a solid medium to extract secondary metabolites. The biological activity of the isolate is presented in Table 3.

The antioxidant activity of the SGI16 crude extract was evaluated using the DPPH test. The results showed that this extract exhibited significant antioxidant power. Indeed, ascorbic acid demonstrated an IC_50_ value of 2.68 ± 0.01 μg/mL, which is close to that of the SGI16 isolate extract, which was 7.24 ± 0.21 μg/mL. The inhibitory activity of the SGI16 extract on BSA denaturation was assessed as a preliminary indicator of its biological activity. The results indicate that both the SGI16 extract and diclofenac inhibited BSA denaturation. Although the inhibitory effect of diclofenac was stronger (IC_50_ = 128.83 ± 0.08 μg mL^−1^) compared to the tested extract (IC_50_ = 492.41 ± 0.47 μg mL^−1^), the extract still showed significant protein denaturation inhibitory activity.

### 3.5. Molecular Characterization of Selected Isolate

The characterization of the actinobacterium SGI16 using *16S rDNA* barcoding identified it as being closely related to the species *Streptomyces yangpuensis,* with the highest similarity percentage of 99.08%, and *Streptomyces flavotricini* and *Streptomyces amritsarensis,* with the same similarity percentage of 98.99. However, in the phylogenetic tree shown in Figure 3, the affiliation of SGI16 is not clearly defined in relation to the closest species: *S. yangpuensis* and *S. amritsarensis*. Therefore, in order to clarify the taxonomic position of the strain SGI16 at the species level, additional tests such as DNA-DNA hybridization and whole genome sequencing should be performed.

## 4. Discussion

Arid and semi-arid regions, characterized by high evaporation rates, tend to accumulate salts in the soil as water evaporates, leaving the dissolved salts behind [47]. Based on the criteria established by the Soil Science Division Staff, the soil in this study is classified as alkaline (pH = 8.07) [48]. This value is close to that reported by previous studies for desert soils [49]. Referring to the salinity scale, which is directly related to electrical conductivity (E.C) as defined by Richards, the studied rhizosphere is not slightly saline (EC = 0.73 dS/m) [26]. Indeed, the scarcity of rainfall is one of the main factors contributing to salinization. During rainfall, water dissolves the salts in the soil and transports them deeper into the ground. Meanwhile, arid and semi-arid regions are characterized by high evaporation rates. When water evaporates from the soil, the salts dissolved in the water remain in the soil [47]. According to the classification by Lee and Hwang, this soil is characterized by low moisture (5.61%) and organic matter content (6.52%) [27]. The rhizosphere is a preferred environment for developing microorganisms due to the availability of nutrients and organic matter from root exudates [50]. In addition, soil characteristics such as pH, salinity, soil water content, soil fertility, etc., are the main factors affecting the distribution and composition of soil microbial communities [51].

The higher counting obtained for SCA medium than ISP_2_ can be attributed to the SCA’s higher concentration of carbon-rich (starch) and nitrogen-rich (casein) substrates as well as the presence of CaCO_3_, which promotes sporulation and consequently increases the number of sporulating actinobacteria. This composition supports the growth of actinobacteria and facilitates their preferential isolation over other bacteria [52]. These researchers confirmed the effectiveness of SCA medium for the selective isolation of actinomycetes from various ecosystems. Moreover, the rhizosphere in this study exhibited a higher abundance of actinomycetes compared to several plant soils in Korea, where the number ranged from 1.17 × 10^6^ to 4.20 × 10^6^ CFU g^−1^ of dry soil [27].

Regarding the enzymatic activities, our results are in line with similar studies. Ranjani et al. reported that several actinomycetes, such as *Streptomyces rubber*, *Thermobifida halotolerant,* and *Thermomonospora* sp., are potent sources of cellulase enzyme [53]. In contrast, Mansour et al. found that none of the actinobacteria they isolated from soil were capable of producing this enzyme [54]. Janatiningrum and Lestari reported that all actinobacteria isolates from the rhizosphere soil of a medicinal *Fiscus deltordea* Jacq. produced amylase [30]. Given that actinobacteria are aerobic bacteria, catalase activity is expected [54]. Catalase released by rhizobacteria can mitigate the effects of reactive oxygen species (ROS) under abiotic stress conditions in plants [55]. Moreover, most isolates from two types of environments (sebkha and palm) in the Algerian Sahara produce lipid-degrading enzymes [22]. Actinobacteria from various ecosystems (rhizosphere, compost, and soils), especially *Streptomyces*, are known to secrete multiple proteases, such as caseinase, gelatinase, peptonization, and milk coagulation [1,56,57]. Additionally, actinobacterial enzymes are important biocatalysts with various applications in industries, including pharmaceuticals, food, paper, textiles, biorefineries, and detergents [8].

The antimicrobial activity that the isolates showed can be attributed to bioactive molecules that inhibit the growth of various test bacteria, mainly through the lysis of their cell walls [30]. Gram-positive tested bacteria were more sensitive to the antibacterial substances produced by the actinobacteria than the Gram-negative ones. This sensitivity has already been noted in similar studies [22,30]. This is often due to their cell membranes’ structural and compositional differences. Indeed, the Gram-negative membrane is composed of phospholipids and glycoproteins, making it much less permeable to lipophilic solutes like antibiotics. However, the Gram-positive membrane consists of an outer peptidoglycan layer that facilitates the penetration of molecules [58]. Recent studies have also reported that actinobacteria from the rhizosphere of various other plants, such as olive, *Junierus excelsa* tree, and bamboo, have been reported for their good antibacterial activity against *S. aureus*, *E. faecalis*, and *B. subtilis* [50,59,60]. The actinomycetes, particularly *Streptomyces* sp., remain the microbes offering the greatest economic and biotechnological benefits, producing the majority of antibiotics used in medicine [50].

Actinomycetes isolated from the rhizosphere of several plants have already shown broad-spectrum antifungal activity against *A. niger*, *Penicillium notatum*, *F. oxysporum*, *Alternaria alternata*, and *C. alibicans* [59,61]. As previously described, the lytic enzymes produced by most of our actinobacteria, such as cellulase, amylase, lipases, and proteases, exert a direct inhibitory effect on fungal pathogens by degrading their cell walls [62].

For many actinomycetes, the production of bioactive molecules is generally more abundant and of better quality when carried out on a solid medium rather than in a liquid one. There are even microorganisms that lose their production capacity when grown in a liquid medium [63]. This difference is attributed to the growth of *Streptomyces* in liquid media, where their hyphae fragment, thereby reducing their ability to produce several bioactive molecules [64]. For the antioxidant activity, the IC_50_ values obtained for SG16 were lower than those found in the literature, indicating that a low amount of the extract is necessary to inhibit the radical (i.e., higher antioxidant activity). For example, *Streptomyces antioxidans* MUSC 164T extracts with a concentration of 2000 µg mL^−1^ showed a DPPH free radical scavenging activity of 18% (which corresponds approximatively to an IC_50_ of 5555 µg mL^−1^) [65]. For *Streptomyces* sp. V002, the DPPH IC_50_ value was 834 µg mL^−1^ [66]. The protein denaturation inhibition results are also consistent with previous findings. Hegazy et al. [67] demonstrated that the pigment extract of *Streptomyces tunisiensis* W4MT573222 effectively inhibited protein denaturation. Their results showed that the extract prevented BSA denaturation, with the inhibitory effect increasing as the pigment concentration increased. At a concentration of 200 µg/mL, the extract achieved an inhibition percentage of 85.9%.

Molecular analysis showed that the SG16 strain belongs to the *Streptomyces* genus. This taxon is commonly found abundant in the rhizosphere of various plants [68], notably in the rhizosphere of several date palm cultivars from the Algerian Sahara [22,23] and Saudi Arabia [69]. *Streptomyces* occupies a prominent position in the field of biotechnology, owing to its rich secondary metabolism, which enables the production of a multitude of bioactive compounds, including several clinically relevant drugs [70]. Indeed, this genus is a prolific source of secondary metabolites such as antibiotics (e.g., streptomycin, gentamicin, tetracycline, chloramphenicol, and erythromycin) [71], antifungals (e.g., piericidin-A1 and nigericin) [72], anticancer agents (e.g., doxorubicin and bleomycin) [73,74], immunosuppressants (e.g., rapamycin) [75], and antivirals (e.g., virantmycin B1) [76]. More than 74% of the antibiotics currently available have been synthesized by this genus [77]. The biomolecules of *Streptomyces* also play a crucial role in agriculture, where they are used as biopesticides and biocontrol agents (such as blasticidin-S and validamycin) [78].

## 5. Conclusions

The present investigation demonstrated that actinobacteria isolated from the rhizosphere of *Phoenix dactylifera* L. in the Algerian Ghardaïa region produce bioactive compounds of biotechnological interest. The findings revealed a wide variety of enzymes and significant antimicrobial, antioxidant, and protein denaturation inhibition activities of the isolates, specifically the isolate *Streptomyces* sp. SGI16. The results offer a promising application in the industrial, pharmaceutical, and agricultural sectors, particularly in the development of new solutions for human and plant health. Additional investigations are needed to further investigate the strain and its extracts. Future research can be directed toward the purification and chemical characterization of the bioactive molecules that can be retrieved from the strains. Biological characterization can be further studied using in vivo models to support their biotechnological use.

## Figures and Tables

**Figure 1 biomolecules-15-00065-f001:**
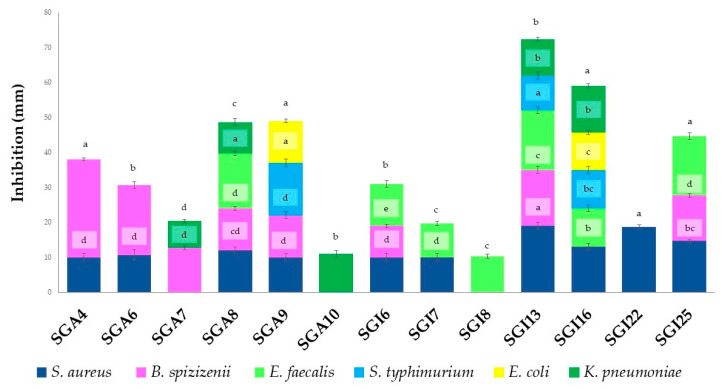
Antibacterial activity of actinobacteria. Vertical bars represent standard error (*n* = 3). Different letters indicate significant differences at *p* ≤ 0.05 according to one-way ANOVA followed by Tukey’s HSD test.

**Figure 2 biomolecules-15-00065-f002:**
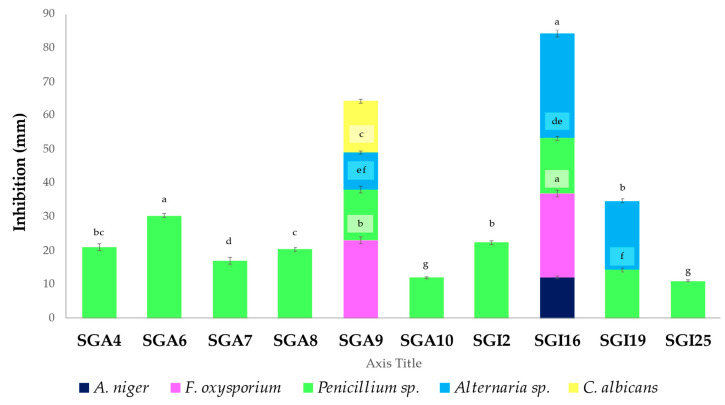
Antifungal activity of actinobacteria. Vertical bars represent standard error (*n* = 3). Different letters indicate significant differences at *p* ≤ 0.05 according to one-way ANOVA followed by Tukey’s HSD test.

**Figure 3 biomolecules-15-00065-f003:**
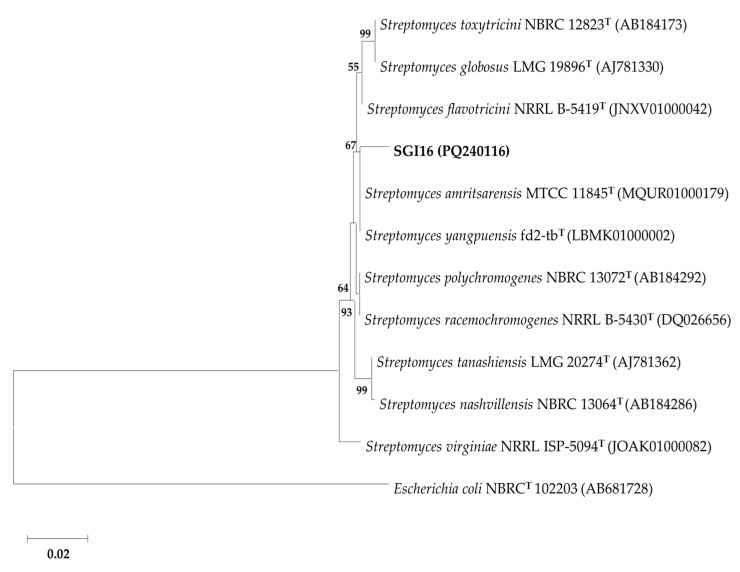
Phylogenetic tree constructed using the neighbor-joining method showing the relationship of isolate SGI16 with the closest species of the genus *Streptomyces*. Bootstrap values above 50% (for 1000 replicates) are indicated. Accession numbers for each sequence are shown in parentheses. The scale bar indicates 0.02 substitutions per nucleotide position. *Escherichia coli* NBRC(T) 102,203 was used as an outgroup.

**Table 1 biomolecules-15-00065-t001:** Physicochemical characteristics of rhizosphere.

pH	Electrical Conductivity (dS/m)	Moisture (%)	Organic Matter (%)
8.07	0.73	5.61	6.52

**Table 2 biomolecules-15-00065-t002:** Enzymatic activity results obtained from actinobacteria isolates. In the table: +, enzymatic activity presence; −, enzymatic activity absence. For Caseinase activity, +, low; ++, medium; +++, high.

Strain	Cellulase	Amylase	Catalase	Esterase	Lipid Metabolism	Caseinase	Gelatinase	Milk Proteins
SGA4	+	+	+	+	Lipoproteinase	+	+	Coagulation
SGA6	+	+	+	+	Lipoproteinase	+	+	Peptonization
SGA7	+	+	+	+	Lecithinase	+++	+	Coagulation
SGA8	+	+	+	+	−	+	−	Peptonization
SGA9	+	+	+	+	Lecithinase	−	+	Coagulation
SGA10	+	-	+	+	−	++	−	Peptonization
SGI2	+	+	+	-	Lipoproteinase	++	+	Peptonization
SGI5	+	+	+	+	Lipoproteinase	++	+	Coagulation
SGI6	+	+	+	-	Lipoproteinase	+	−	Coagulation
SGI7	+	+	+	+	Lipoproteinase	+	−	Peptonization
SGI8	+	+	+	-	Lipoproteinase	+	+	Peptonization
SGI13	+	+	+	+	−	++	+	−
SGI16	+	+	+	+	Lipoproteinase	+++	+	−
SGI18	+	+	+	+	Lipoproteinase	+++	+	Peptonization
SGI19	+	+	+	+	−	+	−	Peptonization
SGI22	+	+	+	+	Lipoproteinase	+	+	Peptonization
SGI24	+	+	+	+	Lipoproteinase	+++	+	Peptonization
SGI25	+	+	+	+	Lipoproteinase	++	+	−

**Table 3 biomolecules-15-00065-t003:** IC_50_ values of the antioxidant activity of SGI16 extract and the reference antioxidant (ascorbic acid) and for BSA denaturation inhibition by SGI16 extract and diclofenac.

Sample	IC_50_ (µg mL^−1^)	IC_50_ (µg mL^−1^)
Isolate SGI16	7.24 ± 0.21 ^a^	492.41 ± 0.47 ^a^
Ascorbic acid	2.68 ± 0.01 ^b^	−
Diclofenac	−	128.83 ± 0.08 ^b^

Values are expressed as mean of three replications ± SD. Different letters indicate significant differences at *p* ≤ 0.05 according to one-way ANOVA followed by Tukey’s HSD test.

## Data Availability

The data supporting this study’s findings are available upon request from the corresponding author.

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
