# Peer review of "Isolation of Actinobacteria from Date Palm Rhizosphere with Enzymatic, Antimicrobial, Antioxidant, and Protein Denaturation Inhibitory Activities"

_biomolecules, 2025, doi:10.3390/biom15010065_

Round 1
Reviewer 1 Report (Previous Reviewer 1)
Comments and Suggestions for Authors
Thank you, for the response to my previuos questions and comments. However I still think, that the scope of the experimental section is insufficient to meet the requirements for publication.
1. Molecular methods should be used for the identification of rhizospheric actinobacteria. This approach could provide valuable insights into the biodiversity of actinobacteria associated with date palms.
2. Alternatively, at least a partial identification of the bioactive compounds could be performed.
Author Response
RESPONSE TO COMMENTS OF REVIEWER 1
Thank you, for the response to my previous questions and comments. However, I still think, that the scope of the experimental section is insufficient to meet the requirements for publication.
R: Thank you for your feedback and for highlighting points that can be improved.
Molecular methods should be used for the identification of rhizospheric actinobacteria. This approach could provide valuable insights into the biodiversity of actinobacteria associated with date palms.
R: Thank you for your suggestion regarding the use of molecular methods for the identification of rhizospheric actinobacteria. We acknowledge that these approaches could enhance the understanding of the biodiversity of actinobacteria associated with date palms.
However, the primary objective of our study was not to investigate the biodiversity of actinobacteria or identify all isolates but rather to focus on the selection and identification of the most performant strain, mainly the one exhibiting the strongest biological activities.
Alternatively, at least a partial identification of the bioactive compounds could be performed.
R: Thank you for your suggestion regarding the identification of bioactive compounds. We acknowledge that even partial identification of the compounds responsible for the observed biological activities would add significant value to our study.
The idea of the study was to select and characterize the best performing actinobacterial strain based on its biological activities. While the identification of bioactive compounds, is relevant, it goes beyond the scope of this study due to methodological constraints and available resources. We appreciate your thoughtful suggestion, which will help guide the next steps in this line of investigation.
Reviewer 2 Report (Previous Reviewer 2)
Comments and Suggestions for Authors
Dear Editor, Dear Authors,
As I indicated once when I reviewed the initial version of this manuscript, the authors could not clame they measured antiinflammatory effect by incubating the extracts with BSA.
Degrading or interacting/modifying BSA is not a prove of an antiinflammatory potential of extract/molecule.
Antiinflammatory effect corresponds to the ability of an extract or molecule to prevent cell/organ/in vivo inflammation and it is measured through assays based on cytokines measurement, NO measurement, or protein expression of markers of inflammation (COX, iNOS, NFkB...)
so this must be corrected and the term "antiinflammatory effect" must be removed.
regards
Author Response
RESPONSE TO COMMENTS OF REVIEWER 2
As I indicated once when I reviewed the initial version of this manuscript, the authors could not clame they measured antiinflammatory effect by incubating the extracts with BSA.
Degrading or interacting/modifying BSA is not a prove of an antiinflammatory potential of extract/molecule. Antiinflammatory effect corresponds to the ability of an extract or molecule to prevent cell/organ/in vivo inflammation and it is measured through assays based on cytokines measurement, NO measurement, or protein expression of markers of inflammation (COX, iNOS, NFkB...). So, this must be corrected and the term "antiinflammatory effect" must be removed.
R: Thank you for your valuable comment. In accordance with your remark, we have revised the manuscript to remove all references linked to "anti-inflammatory effect" and clarified that this method solely evaluates the ability of the extracts to inhibit protein denaturation as a simplified model of biological activity.
The whole manuscript was revised and the term anti-inflammatory was removed and substituted with protein denaturation inhibitory activity. In the discussion part, the paragraph: «Anti-inflammatory results are also consistent with previous findings. Choi et al. [66], who found that, after performing an in vivo test, the extract of an actinobacterium Nocardiopsis sp. 13G027 exerted anti-inflammatory effects on lipopolysaccharide (LPS)-stimulated RAW 264.7 macrophages. This extract significantly inhibited the overexpression of nitric oxide and prostaglandin in LPS-activated RAW 264.7 macrophages» has been replaced with «The protein denaturation inhibition results in this study are consistent with previous work. Hegazy et al. demonstrated that the pigment extract of Streptomyces tunisiensis W4MT573222 effectively inhibited protein denaturation. Their results showed that the extract prevented BSA denaturation, with the inhibitory effect increasing as the pigment concentration increased. At a concentration of 200 µg/mL, the extract achieved an inhibition percentage of 85.9%» to provide a more appropriate discussion. Consequently, the reference [66] (Choi, G.; Kim, G. J.; Choi, H.; Choi, I.W.; Lee, D.S. Anti-Inflammatory and Anti-Fibrotic Activities of Nocardiopsis sp. 13G027 in Lipopolysaccharides-Induced RAW 264.7 Macrophages and Transforming Growth Factor Beta-1-Stimulated Nasal Polyp-Derived Fibroblasts. Microbiol. Biotechnol. Lett. 2021, 49, 543–551, doi: 10.48022/mbl.2109.09016.) has been replaced with: (Hegazy, G. E.; Olama, Z. A.; Abou Elela, G. M.; Ramadan, H. S.; Ibrahim, W. M.; El Badan, D. E. S. Biodiversity and biological applications of marine actinomycetes- Abu-Qir Bay, Mediterranean Sea, Egypt. J. Genet. Eng. Biotechnol. 2023, 21,150, doi: https://doi.org/10.1186/s43141-023-00612-8.).

Round 2
Reviewer 2 Report (Previous Reviewer 2)
Comments and Suggestions for Authors
Dear Editor, Dear Authors
Thank you for addressing my comments
regards
This manuscript is a resubmission of an earlier submission. The following is a list of the peer review reports and author responses from that submission.
Round 1
Reviewer 1 Report
Comments and Suggestions for Authors
The topic of the study is quite interesting and relevant. Actinobacteria, especially less-studied species, or isolated from less-studied environments, are promising source of new bioactive compounds. However, the extent of the work is not sufficient for publication.
Some additional notes:
1. "Sampling was carried out on three different points and collected using the method described by [23] to a depth of 30 cm with a soil corer [24] and stored in refrigerated containers." As I have understand, 3 samples of soils were taken from 3 different points of the date palm field. The rhizosphere is described as a close zone of soil surrounding a plant root, and you used soil corer for sampling, how could you be sure, what your collected soil samples are really from close proximity of plant roots? So, I suppose, you have actinobacterial isolates from soil, but not necessarily from rhizosphere.
2. Molecular methods (for example 16S rDNA analysis) should be performed to confirm if all the isolates are correctly attributed to actinobacteria. Otherwise they should be called bacterial isolates.
3. Please check if all bacterial species and strains are written correctly. For example: "Bacillus subtilis ATCC 6633" (lane 134).is reclassified to another species of genus Bacillus - Bacillus spizizenii (https://bacdive.dsmz.de/strain/1187)
4. Lane 148: - chapter 2.5 should be deleted, because the mentioned methods are described in other chapters.
5. Lane 155: "The crude extracts were filtered through Whatman N° 01 paper (11 μm) and then concentrated to dryness using a rotary evaporator ". What does it mean 11 μm? Is it pore diameter? Is it enough to filter the extract from bacterial cells, or spores?
6. Lane 200: Chapter 3.1 is to short - only one sentence and one table. This chapter could be merged to chapter 3.2.
7. Lane 207: there is no need for Table 2. It has only two numbers and all the information is written in text.
8. It is not correct to count all bacterial colonies as actinobacterial colonies. Bacteria of other phyla could also grow on SCA or ISP agar and some of them could be also resistant to the used antibiotics. And even characterization of morphology of colony is not reliable method for accurate classification.
9. "For Caseinase activty, +, low; ++, medium; +++, high." It is not clear, how do you evaluate the enzymatic activity as low or high. You should specify the diameter of clear halo in mm, for +, ++, and +++ evaluation.
10. Chapter 3.4. I do not understand the results depicted in Figure 1. In the methods part, you have written, what you have measured the diameter of inhibition zone in mm and here you expressed antibacterial activity in percent (%). How did you calculate this? And how did you calculated standart error? Did you repeated your experiment several times? The same I could ask about Figure 2.
11. Description of the identification of the isolate SGI16 should be separated in an other chapter from the characteriasation of its bioactive compounds.
Comments on the Quality of English LanguageThe text should be proofread.
Reviewer 2 Report
Comments and Suggestions for Authors
Dear Editor, Dear Authors
I was invited to evaluate the manuscript « Isolation of actinobacteria from date palm rhizosphere with enzymatic, antimicrobial, antioxidant, and anti-inflammatory activity » by Smati et al.
Here the authors investigated the biological activities of some actinomycetes isolated from the rhizosphere of Phoenix dactylifera L. in the Ghardaia region, Algeria. They isolated 18 actinobacteria and studied them in term of enzymatic and antimicrobial activities. The data provided demonstrated cellulase, catalase, amylase (94%), esterase (84%), lecithinase and lipoproteins (78%), caseinase (94%), gelatinase (72%) activities. In addition, 72% of the isolates exhibited significant antibacterial activity against at least one test bacteria, while 56% demonstrated antifungal activity against at least one test fungi. Then, the authors selected a strain based on enzyme production and antimicrobial activity, the isolate SGI16 and performed secondary metabolite extraction. Molecular identification based on 16S rDNA analysis showed that SGI16 belonged to the genus Streptomyces. The authors concluded that these findings highlight that date palms’ rhizosphere actinobacteria are a valuable source of biomolecules of biotechnological interest.
Overall the study is interesting but needs major and important revisions and could not be accepted without such revisions. Please find below my comments/suggestions/questions :
1- Figure 1 : why the antimicrobial effect is expressed as % inhibition whereas the Mat&Meth indicates an agar diffusion assay ? Data should be expressed as diameter of inhibition and not % of inhibition
2- Same observation for Fig 2
3- why the authors decided to change the read-out/assay for living organisms and crude extracts ? The full organisms were tested for enzymes and antimicrobial, whereas crude extracts were tested for antioxidant and antiinflammatory. The authors must perform same assays both for live organisms and crude extracts. So either do antiox and antiinfl with the living strains and/or perform antimicrobial and enzymes tests with crude extracts.
4- Also very important : I don’t understand the assay used for that by the way : antiinflammatory effect needs inflammation assay using cells and measurement of cytokines production. Testing the effect of the extract on BSA denaturation is not an antiinflammatory assay. This must be corrected.
5- The crude extract must be characterized in term of class of molecules present (at least family of molecules).
6- Minor
« 2.5. Study of antioxidant and anti-inflammatory activities. The SGI16 isolate, which exhibited a broad spectrum of enzymatic, antibacterial and antifungal activities, was selected to investigate its antioxidant and anti-inflammatory potential. » please provide details
« 2.6.2. Evaluation of anti-inflammatory capacity » please put it in italic as other titles of Mat&Meth (same line 179)
regards